# Triglyceride Glucose-Waist Circumference Better Predicts Coronary Calcium Progression Compared with Other Indices of Insulin Resistance: A Longitudinal Observational Study

**DOI:** 10.3390/jcm10010092

**Published:** 2020-12-29

**Authors:** Yun Kyung Cho, Jiwoo Lee, Hwi Seung Kim, Eun Hee Kim, Min Jung Lee, Dong Hyun Yang, Joon-Won Kang, Chang Hee Jung, Joong-Yeol Park, Hong-Kyu Kim, Woo Je Lee

**Affiliations:** 1Department of Internal Medicine, Hallym University Sacred Heart Hospital, Hallym University College of Medicine, Anyang-si 14068, Korea; yukycyk@gmail.com; 2Department of Internal Medicine, Asan Medical Center, University of Ulsan College of Medicine, Seoul 05505, Korea; doublejw825@gmail.com (J.L.); jennyhsk212@gmail.com (H.S.K.); chjung0204@gmail.com (C.H.J.); jypark@amc.seoul.kr (J.-Y.P.); 3Asan Diabetes Center, Asan Medical Center, Seoul 05505, Korea; 4Department of Health Screening and Promotion Center, Asan Medical Center, University of Ulsan College of Medicine, Seoul 05505, Korea; greenk27@hanmail.net (E.H.K.); neige0126@naver.com (M.J.L.); 5Department of Radiology and Research Institute of Radiology, Asan Medical Center, University of Ulsan College of Medicine, Seoul 05505, Korea; donghyun.yang@gmail.com (D.H.Y.); jwonkang@amc.seoul.kr (J.-W.K.)

**Keywords:** coronary artery calcification, insulin resistance, obesity, triglyceride-glucose index

## Abstract

The triglyceride glucose (TyG) index, a product of triglyceride and fasting glucose, is a reliable marker for insulin resistance. We aimed to investigate the association between the TyG-related markers and coronary artery calcification (CAC) progression. We enrolled 1145 asymptomatic participants who underwent repeated CAC score measurements during routine health examinations. Homeostasis model assessment of insulin resistance (HOMA-IR), TyG index, TyG-BMI (body mass index), and TyG-WC (waist circumference) were calculated. Progression of CAC was defined as (1) incident CAC in a CAC-free population, or an (2) increase of ≥2.5 units between the baseline and final square root of the CAC scores in participants with detectable CAC. According to the quartiles of parameters, we stratified the subjects into four groups. The prevalence of progression increased with the TyG-WC quartile (15.0%, 24.1%, 31.0%, and 32.2% for each of the groups; *p* < 0.001). The multivariate-adjusted odds ratio (95% confidence interval) for CAC score progression was 1.66 (1.01–2.77) when the highest and lowest TyG-WC index quartiles were compared. Furthermore, the predictability of TyG-WC for CAC progression was better than the other indices in terms of the area under the curve. The TyG-WC index predicted CAC progression better than other indices and could be a potential marker of future coronary atherosclerosis.

## 1. Introduction

Insulin resistance (IR) connotes an insufficient physiologic response to the effects of insulin, which leads to compensatory hyperinsulinemia [1,2]. IR is a fundamental feature of the pathophysiology of type 2 diabetes and is also related to a wide array of other metabolic derangements, including metabolic syndrome and dyslipidemia [3]. Furthermore, IR also represents a major underlying abnormality driving cardiovascular disease (CVD) [4,5]. 

The hyperinsulinemic-euglycemic clamp (HEC) technique, first described by DeFronzo, is considered as the gold standard measurement of IR [6]. However, it is impractical to apply in real-world clinics due to economic, practical, and ethical limitations [7]. Instead, the homeostasis model assessment of insulin resistance (HOMA-IR) has been widely used for the estimation of insulin resistance in clinical practice and academic field [8]. The TyG index, a product of triglyceride (TG) and fasting plasma glucose (FPG), demonstrated high sensitivity for recognizing IR [7,9,10]. The superiority of TyG in recognizing IR might be achieved as this index incorporated TG along with FPG, both of which are well validated for their roles in IR [7,11,12,13]. Furthermore, several studies found that TyG-related markers that combined obesity indices and the TyG index for IR, such as TyG-BMI (body mass index) or TyG-WC (waist circumference), were more efficient than the TyG index alone [14,15,16].

The coronary artery calcification (CAC) score, which is assessed by noncontrast multi-detector computed tomography (MDCT), reflects the overall coronary plaque burden and a high CAC score is independently and positively associated with future coronary events and prognosis [17]. Moreover, recent studies revealed that CAC progression added incremental value in predicting all-cause mortality over the baseline CAC score, which supports the practical implication of serial assessment of CAC in assessing future cardiovascular risk [18]. As atherosclerosis is a dynamic process, CAC progression could be a better surrogate marker of the activity of atherosclerosis and a predictor for potential CVD events in comparison to baseline CAC [18].

We hypothesized that TyG-related parameters combining obesity indices with the TyG index, such as TyG-BMI or TyG-WC, would be more efficient than the HOMA-IR or TyG index alone in the evaluation of cardiovascular risk. To date, a few studies have shown that TyG-related markers are positively associated with cardiometabolic disease, including CAC [16,19,20]. However, these studies were mostly cross-sectional in design; therefore, the causality between exposure and outcomes could not be evaluated. In light of these findings, we designed our current study to compare HOMA-IR, the TyG index, and TyG-related indices (TyG-BMI and TyG-WC) for prediction of cardiovascular risk by evaluating CAC progression.

## 2. Materials and Methods

### 2.1. Study Population

The study population consisted of 7300 participants who underwent baseline coronary computed tomography angiography (CCTA) using a 64-slice MDCT scanner during health screening assessments at Asan Medical Center (AMC; Seoul, Korea) between January 2007 and June 2011 [21]. Among these 7300 participants, 1591 participant underwent repeated CCTA through December 2014 [21]. In accordance with the ethical guidelines of the Declaration of Helsinki and Korea Good Clinical Practice, this study was approved by the institutional review board of the Asan Medical Center (AMC; 2020-0343). Consent from individual subjects was not needed as anonymous archival data without identifying information was used [22].

We analyzed the demographic and biochemical data obtained from in-person follow-up examinations after the baseline examinations [21]. Each participant completed a questionnaire that listed a history of previous medical and/or surgical diseases, medications, and drinking and smoking habits. Drinking habits were categorized in terms of frequency per week (i.e., more than 2 times/week (moderate drinker)), smoking habits were classified as noncurrent or current, and exercise habits were described as frequency per week (i.e., more than 3 times/week (physically active)) [21]. The history of CVD was defined on each participant’s history of clinician-diagnosed angina, myocardial infarction, and/or cerebrovascular accidents [21]. Participants with a fasting plasma glucose (FPG) level of ≥7.0 mmol/L and/or HbA1c level 6.5% and/or the prescription history of antidiabetic medications on a self-report questionnaire were defined to have diabetes [21,23]. Hypertension was defined as systolic and/or diastolic blood pressure (BP) ≥140/90 mmHg or using antihypertensive medications [21].

We excluded participants with a history of CVD at the baseline examination (*n* = 95), prescription of statins (*n* = 238), a history of percutaneous coronary intervention (*n* = 8), coronary arterial bypass surgery (*n* = 3) after the baseline examination, or missing data (*n* = 109) [22]. We also excluded participants who were younger than 20 years or older than 79 years (*n* = 3). Some participants met more than 2 exclusion criteria. After excluding ineligible subjects, 1145 participants with a mean age of 54.2 ± 7.6 years were included as our final cohort in the analysis (Figure 1). 

### 2.2. Clinical and Laboratory Measurements

Height and weight were obtained while the participants wore light clothing without shoes. Body mass index (BMI) was calculated as weight in kilograms divided by the square of the height in meters [21]. Waist circumference (WC) (in cm) was measured in the horizontal plane midway between the lowest rib and the iliac crest at the end of a normal expiration [21]. Blood pressure (BP) was measured on the right arm after resting confortably for 5 min using an automatic manometer with an appropriate cuff size [21]. After overnight fasting, early morning blood samples were drawn from the antecubital vein into vacuum tubes and subsequently analyzed by the central, certified laboratory at AMC [21]. Measurements included the concentrations of HbA1c, fasting glucose, lipid parameters, and liver enzymes and high-sensitivity C-reactive protein (hsCRP) [21].

Fasting total cholesterol (TC), high-density lipoprotein cholesterol (HDL-C), triglycerides (TG), uric acid, aspartate aminotransferase (AST), and alanine aminotransferase (ALT) were measured using the enzymatic colorimetric method on a Toshiba 200FR Neo analyzer (Toshiba Medical System Co., Ltd., Tokyo, Japan) [21]. Low-density lipoprotein cholesterol (LDL-C) was directly measured using the enzymatic colorimetric method on a Toshiba 200FR Neo analyzer (Toshiba Medical System Co., Ltd., Tokyo, Japan) [21]. Gamma-glutamyl transferase (GGT) was measured using the L-g-glutamyl-p-nitroanilide method (Toshiba). HsCRP and FPG were measured using the immunoturbidimetric method (Toshiba) and the enzymatic colorimetric method on a Toshiba 200 FR auto-analyzer (Toshiba), respectively. Ion-exchange high-performance liquid chromatography (Bio-Rad Laboratories, Inc., Hercules, CA) was used to measure the HbA1c levels [21]. All enzyme activities were measured at 37 °C [21].

Insulin resistance was measured using the HOMA-IR according to the following equation: (fasting insulin [µIU/mL] × fasting glucose [mg/dL])/405 [24]. Other TyG-related markers were calculated using the following formulae: TyG index = Ln [TG (mg/dL) × FPG (mg/dL)/2]; TyG-BMI = TyG index × BMI (kg/m^2^); and TyG-WC = TyG index × WC (cm) [20].

### 2.3. Use of MDCT to Assess the CAC Score

MDCT examinations were carried out using either a 64-slice, single-source (LightSpeed VCT; GE, Milwaukee, WI) or a 64-slice, dual-source (Somatom Definition; Siemens, Erlangen, Germany) scanner [21,22,25,26]. The CAC score was calculated using an automated software program using the Agatston scoring method [21,22,27], and participants were categorized according to the cut-off points used by Greenland et al. (i.e., none, 0; mild, 1–100; moderate, 101–300; severe, >300) [21,22,28].

Progression of CAC was defined as (1) incident CAC, which indicates a baseline Agatston score of zero converting to detectable CAC at a follow-up examination in a population free of CAC at baseline [22,29,30], or (2) an increase of ≥2.5 units between the baseline and final square root of CAC scores participants who had detectable CAC at the baseline examination [21,31,32]. To avoid the dependence of residual interscan variability, we performed square root transformation of the CAC score in the determination of CAC progression. Using the data published by Hokanson et al., progressors were defined as individuals with a difference of ≥2.5 units between the baseline and final square root of their CAC scores (i.e., the “SQRT method” (the square root-transformed difference)) [18,22,31,32]. In other words, as a change of less than 2.5 units between the baseline and final square root of the CAC score might be within the margin of error for the estimation of the CAC score using MDCT and thus was attributed to interscan variability, such participants were categorized as non-progressors [18,22,31,32].

### 2.4. Statistical Analysis

Continuous variables with regular distributions are presented as the mean ± standard deviation, whereas continuous variables with skewed distributions are expressed as the median (and interquartile range). Categorical variables are expressed as the percentage. To evaluate the continuous variables, one-way analysis of variance (ANOVA) with Scheffe’s method was performed and the chi-squared test was performed to compare the categorical variables among subgroups. The demographic and biochemical characteristics of subgroups categorized by the CAC score progression were compared using Student’s *t*-test for normally distributed continuous variables or Mann–Whitney *U* test for continuous variables not normally distributed, and the chi-squared test for categorical variables. Logistic regression analysis was performed to calculate the odds ratios (ORs) and 95% confidence intervals (CIs) of the subgroups for the prediction of CAC progression. To assess the utility of parameters for the prediction of CAC score progression, we conducted receiver operating characteristics (ROC) curves, calculated the areas under the curve (AUC), and compared AUCs by DeLong method [33]. The AUCs were determined using MedCalc version 11.12.0 for Windows (MedCalc Software, Mariakerke, Belgium). All statistical analyses, except for the ROC curve analysis, were performed using SPSS version 20.0 for Windows (SPSS Inc., Chicago, IL, USA). In our current analyses, *p* < 0.05 was considered statistically significant.

## 3. Results

### 3.1. Clinical and Biochemical Characteristics of the Study Participants

The baseline biochemical and clinical characteristics of the study subjects according to CAC progression are shown in Table 1. Among the 1145 participants, the prevalence of CAC progression was 25.6% (*n* = 293). The mean participant age was 54.2 ± 7.6 years, and progressors were more likely to be older than non-progressors. Compared with non-progressors, progressors demonstrated a higher BMI, WC, and systolic and diastolic BP. In addition, progressors were more likely to be male, current smokers, and frequent drinkers. Progressors had a less favorable risk profile, which included a higher prevalence of hypertension and diabetes and higher levels of FPG, HbA1c, uric acid, AST, ALT, and GGT. The level of HDL-C was significantly lower in progressors than in the non-progressors. Most metabolic parameters, including TyG, TyG-BMI, and TyG-WC, were higher in progressors compared to non-progressors; however, there was no difference in HOMA-IR between the groups. Progressors had higher baseline CAC scores and were followed up for longer periods than non-progressors.

### 3.2. Relationship of HOMA-IR and TyG-Related Markers with CAC Score Progression

When participants were divided into quartiles of metabolic parameters, the proportions of subjects with CAC progression showed a tendency to increase with higher metabolic parameters (*p* = 0.031, *p* = 0.007, *p* < 0.001, and *p* < 0.001 for the linear trend) (Figure 2). In particular, there was a graded association between CAC score progression and HOMA-IR and TyG-WC. 

Figure 3 shows the annualized change in the CAC score according to the quartiles of metabolic parameters. The quartiles of TyG, TyG-BMI, and TyG-WC were significantly and positively associated with the annualized change of CAC score. In particular, the annualized change of CAC score was linearly increased when the quartiles of TyG-BMI or TyG-WC were applied.

Next, the ORs of CAC progression were calculated according to the quartiles of metabolic parameters. When compared with the participants with TyG-BMI in the first quartile, the participants with TyG-BMI in the fourth quartile were 1.62 times (95% CI 1.00–2.62) more likely to exhibit CAC score progression, even after adjustment for confounding variables (Table 2 and Figure 4). When the TyG-WC quartiles were applied, the risk of CAC score progression was significantly increased in the third and fourth quartiles in comparison to the first quartile, with higher ORs (OR 1.64; 95% CI 1.01–2.66 and OR 1.66; 95% CI 1.01–2.77, respectively) (Table 2 and Figure 4). Using TyG quartiles, the participants with TyG in the second quartiles had a higher risk of CAC progression (OR 1.65; 95% CI 1.06–2.57) compared to the first quartile, however, the risk was not increased in the third and fourth quartile groups. The association of HOMA-IR and CAC progression was not statistically meaningful after full adjustment of other covariates.

### 3.3. Comparison of HOMA-IR and TyG-Related Markers for the Prediction of CAC Score Progression

The metabolic parameters achieved moderate prognostic performance for CAC progression. The highest AUC was demonstrated by TyG-WC (AUC = 0.600), followed by TyG-BMI (AUC = 0.583), TyG (AUC = 0.557), and HOMA-IR (AUC = 0.543) (Figure 5 and Table 3). TyG-WC had significantly higher AUC values compared to the HOMA-IR and TyG index (*p* = 0.010 vs. HOMA-IR, and *p* = 0.011 vs. TyG).

## 4. Discussion

In the present longitudinal study using a large health-screening cohort, we observed that individuals with high TyG-related indices are more likely to experience CAC progression. Among these indices, TyG-WC showed the strongest association with CAC progression, a marker that predicts coronary artery disease and patient prognosis. In particular, our analyses revealed that TyG-WC was positively associated with CAC score progression, independently of conventional cardiovascular risk factors. Participants with TyG-WC in the highest quartile were 1.66 times more likely to suffer from CAC progression after adjustment for confounding variables compared with those with TyG-WC in the lowest quartile. According to ROC analysis, TyG-WC was the most reliable predictor of CAC progression among the parameters we evaluated. This positive association between TyG-WC and CAC progression implies adverse cardiovascular outcomes in individuals with a high TyG-WC, considering the predictive value of CAC progression for future adverse cardiovascular outcomes.

Coronary calcification is a surrogate marker of atherosclerosis, and the CAC score is a representative anatomic measure of overall coronary plaque burden [34]. It has been reported that CAC could provide independent incremental information in addition to traditional risk factors in the prediction of adverse cardiac events and mortality in symptomatic patients [34,35] as well as asymptomatic populations [36,37]. Moreover, more recent evidence supports the progression of CAC as a predictor of future cardiovascular events [18,38,39,40]. It has been reported that CAC progression is associated with increases in all-cause mortality as well as incident, hard, and total coronary heart disease events in large prospective cohort studies [18,39]. In our present study, we assessed CAC score progression using serial CT scans, which were performed at an average of 3 years apart. Considering the dynamic nature of atherosclerosis, evaluating CAC progression has clinical value in assessing atherosclerosis progression and future cardiovascular risk.

IR is one of the most important mechanisms in the development and advancement of cardiovascular disease by promoting atherogenesis and plaque progression [20,41,42,43]. HOMA-IR is a representative surrogate marker for insulin resistance with great utility in clinical and laboratory settings since the use of HEC, the gold standard for estimating insulin resistance, is limited due to the inconvenience and high cost [20,44,45]. However, conflicting results have been reported by previous reports on the association between HOMA-IR and CAC progression. Two cohort studies performed in Japan and South Korea showed that a higher HOMA-IR was, independently of other cardiovascular risk factors, associated with an increase in CAC score [46,47]. In contrast, Lee et al. [48] observed no association of HOMA-IR with CAC progression in a large, community-based prospective study. Similarly, a large cohort study including 5464 participants not receiving hypoglycemic therapy from the Multi-Ethnic Study of Atherosclerosis (MESA) revealed that HOMA-IR was not predictive of CAC progression after adjustment for metabolic syndrome components [49]. The conflicting data regarding the association of HOMA-IR and CAC progression could depend on the different adjustment of the models, the difference in the research participants, and the different definitions of CAC progression. In the present analysis, similarly to the latter two studies, we found that although groups with a higher HOMA-IR included more progressors (Figure 2), the HOMA-IR was not significantly associated with CAC progression after adjustment for other conventional risk factors (Table 2 and Figure 4). 

In the present study, we evaluated TyG and TyG-related parameters (i.e., TyG-BMI and TyG-WC) as predictors of CAC progression. The TyG index proposed by Guerrero-Romero et al. has shown high sensitivity and specificity in the identification of IR in several studies, and therefore it could be useful for detection of subjects with impaired insulin sensitivity [7,9,10,14,50]. Indeed, hepatic triglyceride content is a strong determinant of hepatic insulin resistance [51,52] and the intramyocellular triglyceride content of the muscle insulin resistance [53], supporting the close relationship of triglyceride with insulin resistance [14]. Furthermore, it has been reported that an elevated TyG index is related to poor cardiovascular outcomes, such as a higher prevalence of symptomatic coronary artery disease and more major adverse cardiac and cerebrovascular events [50,54,55,56]. Recently, Park et al. reported an independent association of elevated TyG index level with CAC progression in Korean adults [57]. Similarly, in the present study, we observed a positive association between a higher TyG index and CAC progression. However, the OR of CAC progression was not linearly increased by TyG quartiles; the ORs for CAC progression were not significantly increased in the third and fourth quartiles, showing the highest OR in the second quartile after adjustment for other variables (Table 2 and Figure 4).

In contrast, TyG-BMI and TyG-WC, which are combined measurements of TyG and obesity indices, demonstrated a graded association with CAC progression (Table 2 and Figure 4). In addition to insulin resistance, obesity per se has been shown to be positively associated with cardiovascular disease. Accordingly, we hypothesized that the performance of the TyG index combined with obesity indices (i.e., TyG-BMI and TyG-WC) to predict CAC progression would be better than the performance of HOMA-IR or TyG index alone, which was subsequently supported by our study findings. Our results showed the discriminative ability of TyG-WC for CAC progression; specifically, TyG-WC performed better than the other parameters with the highest ORs for CAC progression in the third and fourth quartiles (Table 2 and Figure 4), and the largest AUC (Figure 5 and Table 3). Recently, Kim et al. showed that TyG-WC and TyG-BMI predicts CAC better than other indices of insulin resistance (i.e., HOMA-IR and TyG index) [20]. Similarly, in our whole study cohort who underwent coronary CT (Appendix A), TyG-related indices including TyG-WC had the strong association with CAC, as well as CAC progression. At baseline health examinations, the prevalence of CAC was 33.7%, and the proportions of subjects with CAC showed a tendency to increase with higher metabolic parameters (Appendix A and Appendix A). Multivariate logistic regression analysis showed that participants with TyG-WC in the highest quartile were 1.91 times more likely to have CAC after adjustment for other covariates, compared with those with TyG-WC in the lowest quartile (OR, 1.91; 95% CI, 1.45–2.50; Appendix A and Appendix A). The ROC curve for the presence of CAC showed that TyG-WC index had a higher AUC than other indices (AUC_HOMA-IR_ = 0.564, AUC_TyG_ = 0.592, AUC_TyG-BMI_ = 0.599, and AUC_TyG-WC_ = 0.622; Appendix A and Appendix A), which is in line with the previous finding by Kim et al. [20]. However, as these results were derived from cross-sectional examinations, a causal relationship between the parameters and CAC could not be derived. Our present study provides additional evidence supporting the use of TyG-related indices reflecting adiposity for predicting cardiovascular disease by assessing CAC progression in a longitudinal observational study. 

Our present study had some limitations. First, our analyses involved only Korean participants from a single center; therefore, the results might not be applicable to other populations. Considering the variability of TG levels according to ethnicity, further studies will be necessary to test whether TyG, TyG-BMI, and TyG-WC are suitable markers for CAC progression in other populations. Second, the present study compared TyG-related parameters with HOMA-IR, not the HEC; therefore, we could not provide evidence supporting the advantages of TyG-related indices over HEC, the gold standard measurement of insulin sensitivity. However, as the glucose clamp is time-consuming, costly, and complex, it is difficult to apply in clinic, which favors the use of TyG-related indices in the clinical setting rather than HEC. Third, the adjustment we used in the logistic regression analyses might affect the results; therefore, more or less covariates may have yielded different results. Fourth, since there is no consensus on the definition of CAC progression, the definition used in our analyses might be arbitrary. The absolute changes in the CAC scores between baseline and follow-up [38,58], or the mean changes in square root transformed score [18,59] have been used in previous studies. However, it has been shown that the best CAC progression model predicting mortality is the SQRT method, which we decided to use in the present study, and a SQRT difference of 2.5 provides the best fit for the data [18]. Finally, although CAC progression is a well-known surrogate marker of cardiovascular risk, further prospective studies are needed to warrant the predictive value of TyG-related markers for the hard cardiovascular outcomes. Despite these above limitations, this is the first study to longitudinally assess TyG–obesity combined indices as predictors of cardiovascular risk as evaluated by CAC progression across a large number of participants.

## 5. Conclusions

In our study, on an asymptomatic middle-aged population, TyG-related indices were significantly correlated with CAC progression. Among those indices, TyG-WC was the most reliable marker for predicting CAC progression in healthy Koreans. Given the fact that CAC progression is a surrogate marker of cardiovascular risk, TyG-WC could be predictive of future coronary disease and patient prognosis. Considering that the TyG-WC index can easily be calculated as the required values can be obtained from simple anthropometric measurements and routine laboratory tests, we recommend the application of TyG-WC in cardiovascular risk estimation in real clinical practice and epidemiologic surveys.

## Figures and Tables

**Figure 1 jcm-10-00092-f001:**
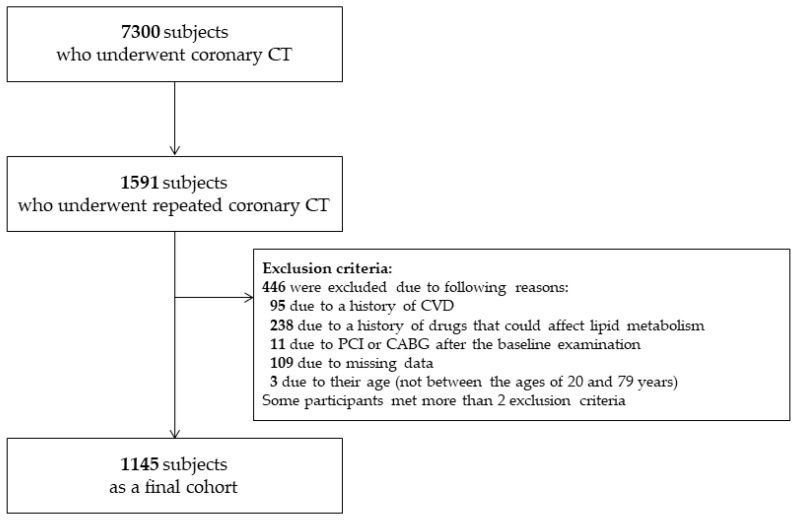
Study population. CABG, coronary artery bypass grafting, CVD, cardiovascular disease; PCI, percutaneous coronary intervention.

**Figure 2 jcm-10-00092-f002:**
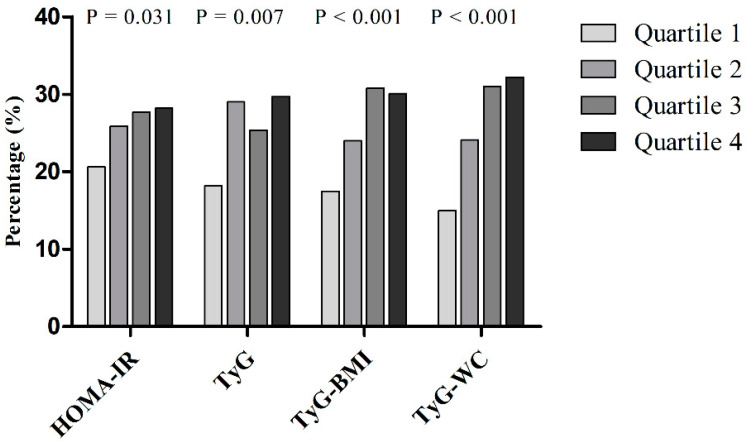
Proportion of progressors by their CAC scores according to the quartiles of homeostasis model assessment of insulin resistance (HOMA-IR), triglyceride glucose (TyG), TyG-BMI (body mass index), and TyG-WC (waist circumference) in the final cohort including non-progressors and progressors (*n* = 1145). The proportion of progressors was compared using the chi-squared test.

**Figure 3 jcm-10-00092-f003:**
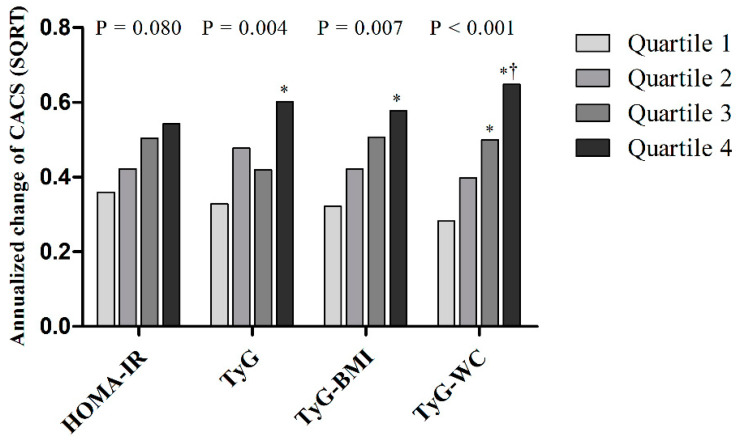
Annualized change of CAC according to quartiles of each parameter in the final cohort including non-progressors and progressors (*n* = 1145). The *p*-values indicate overall *p*-value of the ANOVA and the symbols indicate results from the post hoc test of Scheffe’s method (* *p* < 0.05 vs. Quartile 1, † *p* < 0.05 vs. Quartile 2).

**Figure 4 jcm-10-00092-f004:**
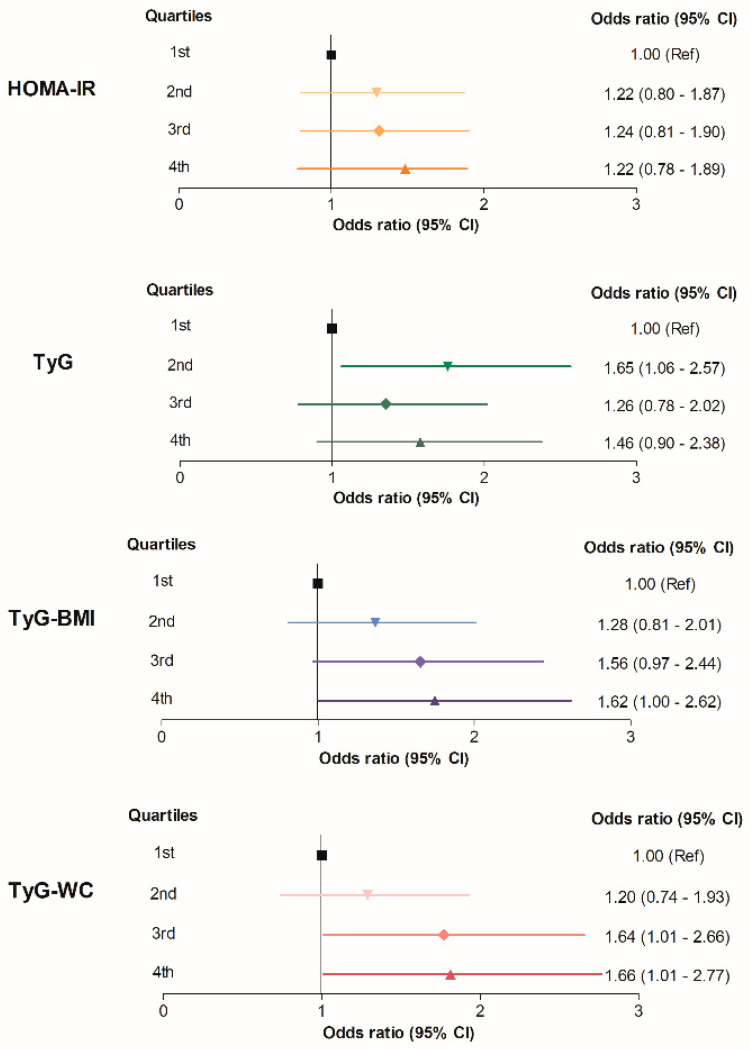
Summarized figure for CAC progression risk according to the quartiles of each parameter in the final cohort including non-progressors and progressors (*n* = 1145). The ORs (95% CIs) are adjusted for age, sex, systolic BP, LDL-C, HDL-C, smoking, drinking, exercise habits, baseline CAC score, and follow-up interval.

**Figure 5 jcm-10-00092-f005:**
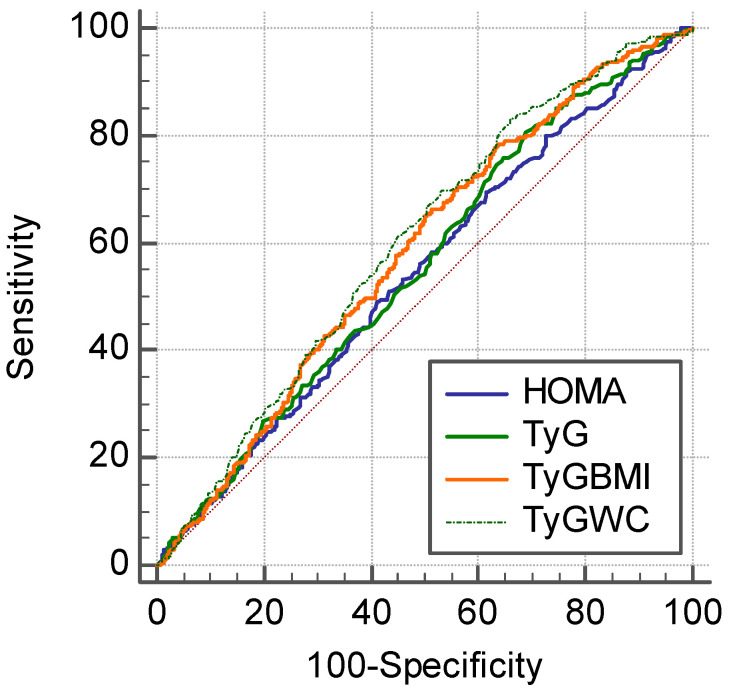
Receiver operating characteristic (ROC) curve of metabolic parameters for the CAC progression in the final cohort including non-progressors and progressors (*n* = 1145).

**Table 1 jcm-10-00092-t001:** Baseline clinical and biochemical characteristics according to the coronary artery calcification (CAC) score progression in the final cohort (*n* = 1145).

	Total	Non-Progressor	Progressor	*p*
*n* (%)	1145	852 (74.4)	293 (25.6)	
Age (years)	54.2 ± 7.6	53.5 ± 7.2	56.3 ± 8.1	<0.001
Sex (male, %)	81.7	78.2	91.8	<0.001
Body mass index (kg/m^2^)	25.0 ± 3.0	24.8 ± 3.1	25.5 ± 2.6	0.002
Waist circumference (cm)	87.1 ± 8.3	86.5 ± 8.4	89.2 ± 7.4	<0.001
Systolic BP (mmHg)	119.0 ± 12.6	118.0 ± 12.1	121.8 ± 13.4	<0.001
Diastolic BP (mmHg)	76.2 ± 10.4	75.6 ± 10.2	78.0 ± 10.9	0.001
Current smoker (%)	27.4	25.2	33.8	0.006
Moderate drinker (%)	52.1	50.2	57.3	0.042
Physically active (%)	44.3	42.7	48.8	0.076
Family history of diabetes (%)	24.4	23.9	25.6	0.581
Diabetes (%)	13.5	11.5	19.5	0.001
Hypertension (%)	33.0	28.8	45.4	<0.001
FPG (mg/dL)	104.9 ± 19.1	103.9 ± 18.7	108.0 ± 19.9	0.002
HbA1c (%)	5.7 ± 0.7	5.6 ± 0.7	5.8 ± 0.9	0.002
HbA1c (mmol/mol)	38.3 ± 8.2	37.9 ± 7.7	39.6 ± 9.3	0.002
Total cholesterol (mg/dL)	197.8 ± 32.6	198.2 ± 32.3	196.4 ± 33.5	0.410
TG (mg/dL)	133.3 ± 77.3	131.4 ± 76.8	138.9 ± 78.6	0.154
LDL-C (mg/dL)	125.3 ± 28.9	125.3 ± 28.8	125.3 ± 29.2	0.980
HDL-C (mg/dL)	51.6 ± 13.2	52.2 ± 13.7	49.9 ± 11.5	0.004
Uric acid (mg/dL)	5.8 ± 1.4	5.7 ± 1.4	6.0 ± 1.3	0.001
AST (U/L)	25 (21–31)	25 (21–31)	27 (23–34)	0.001
ALT (U/L)	23 (17–32)	22 (17–31)	24 (19–35)	0.001
GGT (U/L)	25 (16–40)	24 (16–38)	30 (20–44)	<0.001
hsCRP (mg/L)	0.6 (0.3–1.3)	0.6 (0.3–1.3)	0.7 (0.4–1.4)	0.079
HOMA-IR	2.1 ± 1.5	2.1 ± 1.5	2.3 ± 1.5	0.116
TyG index	8.7 ± 0.6	8.7 ± 0.6	8.8 ± 0.5	0.005
TyG-BMI	218.3 ± 33.4	216.2 ± 34.6	224.2 ± 29.0	<0.001
TyG-WC	760.5 ± 99.4	752.3 ± 101.4	784.2 ± 89.1	<0.001
Baseline CAC score	0 (0–22)	0 (0–10)	12 (0–99)	<0.001
Last follow-up CAC score	1 (0–50)	0 (0–16)	66 (10–248)	<0.001
Baseline CAC score category				<0.001
0 (%)	57.2	65.1	35.5	
>0 (%)	42.8	34.9	64.5	
Follow-up interval (years)	3.0 (2.1–3.9)	2.9 (2.0–3.8)	3.1 (2.5–4.0)	<0.001

BP, blood pressure; FPG, fasting plasma glucose; HbA1c, hemoglobin A1c; TG, triglyceride; LDL-C, low-density lipoprotein cholesterol; HDL-C, high-density lipoprotein cholesterol; AST, aspartate aminotransferase; ALT, alanine aminotransferase; GGT, gamma-glutamyl transferase; hsCRP, high-sensitivity C-reactive protein; CAC, coronary artery calcification. *p*-value shows comparison between non-progressor and progressor groups. Continuous variables with normal distributions are expressed as the mean ± standard deviation, whereas continuous variables with skewed distributions are expressed as the median (and interquartile range). Categorical variables are expressed as the percentage. The characteristics were compared using Student’s *t*-test for normally distributed continuous variables or Mann–Whitney *U* test for continuous variables not normally distributed, and the chi-squared test for categorical variables.

**Table 2 jcm-10-00092-t002:** CAC progression according to the quartiles of each parameter in the final cohort including non-progressors and progressors (*n* = 1145).

Parameter	*n*	OR (95% CI)			
		Unadjusted	Model 1	Model 2	Model 3
HOMA-IR					
First quartile	287	1.00 (Ref)	1.00 (Ref)	1.00 (Ref)	1.00 (Ref)
Second quartile	286	1.35 (0.91–1.99)	1.33 (0.89–1.99)	1.27 (0.84–1.92)	1.22 (0.80–1.87)
Third quartile	285	1.48 (1.01–2.18)	1.36 (0.91–2.03)	1.24 (0.82–1.88)	1.24 (0.81–1.90)
Fourth quartile	287	1.52 (1.03–2.23)	1.41 (0.95–2.10)	1.28 (0.83–1.96)	1.22 (0.78–1.89)
TyG					
First quartile	286	1.00 (Ref)	1.00 (Ref)	1.00 (Ref)	1.00 (Ref)
Second quartile	286	1.84 (1.24–2.73)	1.67 (1.11–2.52)	1.52 (0.99–2.33)	1.65 (1.06–2.57)
Third quartile	287	1.54 (1.03–2.29)	1.36 (0.90–2.08)	1.20 (0.76–1.91)	1.26 (0.78–2.02)
Fourth quartile	286	1.90 (1.28–2.82)	1.72 (1.14–2.61)	1.43 (0.89–2.30)	1.46 (0.90–2.38)
TyG-BMI					
First quartile	286	1.00 (Ref)	1.00 (Ref)	1.00 (Ref)	1.00 (Ref)
Second quartile	287	1.49 (0.99–2.25)	1.26 (0.83–1.93)	1.25 (0.80–1.93)	1.28 (0.81–2.01)
Third quartile	286	2.10 (1.41–3.11)	1.71 (1.13–2.59)	1.55 (1.00–2.45)	1.56 (0.97–2.44)
Fourth quartile	286	2.03 (1.37–3.02)	1.83 (1.21–2.79)	1.68 (1.05–2.69)	1.62 (1.00–2.62)
TyG-WC					
First quartile	286	1.00 (Ref)	1.00 (Ref)	1.00 (Ref)	1.00 (Ref)
Second quartile	286	1.80 (1.18–2.74)	1.33 (0.86–2.07)	1.27 (0.80–2.01)	1.20 (0.74–1.93)
Third quartile	287	2.54 (1.69–3.83)	1.92 (1.24–2.96)	1.78 (1.11–2.86)	1.64 (1.01–2.66)
Fourth quartile	286	2.68 (1.78–4.03)	1.96 (1.27–3.03)	1.80 (1.10–2.94)	1.66 (1.01–2.77)

Model 1 was adjusted for age and sex. Model 2 was adjusted for the variables included in model 1, plus systolic BP, LDL-C, HDL-C, smoking, drinking, and exercise habits. Model 3 was adjusted for the variables included in model 2, plus baseline CAC score and follow-up interval. Logistic regression analysis was performed to calculate the odds ratios (ORs) and 95% confidence intervals (CIs) of the subgroups for the prediction of CAC progression.

**Table 3 jcm-10-00092-t003:** Areas under the receiver operating characteristic curves of each parameter for CAC progression in the final cohort including non-progressors and progressors (*n* = 1145).

**Parameter**	**AUC**	**Standard Error**
HOMA-IR	0.543	0.0193
TyG	0.557	0.0189
TyG-BMI	0.583	0.0186
TyG-WC	0.600	0.0184
**Comparison ***	**Difference AUC**	***p*-Value ***
TyG-WC vs. HOMA-IR	0.057	0.010
TyG-WC vs. TyG	0.043	0.011
TyG-WC vs. TyG-BMI	0.017	0.202
TyG-BMI vs. HOMA-IR	0.040	0.176
TyG-BMI vs. TyG	0.026	0.527
TyG vs. HOMA-IR	0.014	1.000

The difference of prediction performance between the parameters were presented in the ROC curve (AUC) between the models. * Comparisons were adjusted for multiple comparisons using Bonferroni correction. AUC = area under receiver operating characteristic (ROC) curves; CI = confidence interval.

## Data Availability

The data presented in this study are available on request from the corresponding author. The data are not publicly available due to ethical restrictions.

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
