# Peer review of "Triglyceride Glucose-Waist Circumference Better Predicts Coronary Calcium Progression Compared with Other Indices of Insulin Resistance: A Longitudinal Observational Study"

_jcm, 2020, doi:10.3390/jcm10010092_

Round 1

Reviewer 1 Report

Cho et al.'s manuscript using a large prospective Korean cohort examines the predictors of atherosclerosis progression as measured by CAC score. The group identifies Triglyceride-glucose homeostasis and adiposity related measurements such as TyG-BMI and TyG-WC to correlate with changes in CAC score better than HOMA-IR.  The findings are relevant and important. The studies are rigorous, the manuscript is well written, and the data is clearly presented. There are minor comments that will enhance the clarity of the manuscript.

  • It is unclear why 7,300 participants are mentioned when the study focuses on a subcohort of 1591 participants with repeated CAC measurements. The cross-sectional examination of CAC scores as 0 and >0 in the whole cohort would add additional information, and comparing the results to other studies such as referenced in #20 will enhance the overall value of the identification of TyG-BMI and TyG-WC as strong players in CAD risk prediction
  • A flow chart of the cohort that ends up to be in the analytical sample size will add clarity and a visual summary to Lines 90-95.
  • In Methods, please specify if LDL-C is directly measured or calculated.
  • Models 2 and 3 in table 2 adjust for LDL-C, which is not different between non-progressors and progressors. On the other hand, HDL-C is statistically lower in progressors, which is following the current understanding of the HDL-C’s role in cardiovascular disease. The authors could consider accounting for HDL-C in the regression model, strengthening the model and its biological significance.
  • Please specify if point estimates presented in Figure 3 are adjusted.
  • The pair-wise comparisons of the ROCs (presented in Table 3) should be adjusted for multiple comparisons.

Author Response

On behalf of the authors, I would like to thank you and the reviewers for the valuable and helpful comments on our submitted manuscript, which helped improve this manuscript substantially.

We have carefully reviewed the comments and have revised the manuscript accordingly. Our responses are provided in a point-by-point manner, and changes to the manuscript are shown in red color.

Please see the attachment for our point-by-point responses to your comment.

Reviewer 2 Report

Yun Kyung Cho and colleagues investigated the application of metabolic and obesity indices in order to predict coronary calcium progression in asymptomatic subjects. The topic of the article is interesting and the results suggest that triglyceride glucose – waist circumference (TyG-WC) could predict the coronary calcium progression. Nevertheless, several issues are raised.

Major points

  1. In one hand, the authors adjusted the used models (as in model 3, Table 2) for several parameters including sex, age, smoking status, baseline CAC score, etc. On the other hand, very important confounding parameters like BMI, waist circumference, diabetes status, fasting plasma glucose concentration, HbA1c were not included in the model. First, these parameters show significant differences between progressors and non-progressors (Table 1); second these parameters are known to influence glucose and insulin metabolism in the human physiology. In order to assess whether TyG-WC could indeed predict CAC progression, one should adjust the model also for these parameters.
  2. The authors should check and if applicable should correct the calculation of HOMA-IR (line 116). The current formula of calculating HOMA-IR is:

(fasting insulin [µIU/ml] x fasting glucose [mg/dl]) /405) or

(fasting insulin [µIU/ml] x fasting glucose [mmol/l]) /22.5)

  1. The authors analyzed data of progressors and non-progressors (Table 1). In the later sections the authors show figures and tables, however for the reviewer is not clear whether these data were collected only from the progressor subjects or also include data from non-progressors. It would be useful to add this information for each figure and table and also include the number of patients, whose data were collected.
  2. In figure 2 p-values are shown as numbers above the figure and symbols (like stars and cross) are also applied. The reviewer assumes that the authors applied at least two different statistical tests. Therefore all statistical tests should be identified in the figure legends to show the reader which p-values belong to which statistical tests.
  3. The authors show some data as Table 3 for the ROC curves including AUC, however it would be useful to see the original ROC curves as figures.
  4. In the discussion (line 255), the authors summarize literature data regarding the association between HOMA-IR and CAC progression. The authors claim, that the previous data are conflicting, since some studies observed a significant association between HOMA-IR and CAC progression, however some did not. As it was mentioned by the first comment, choosing all relevant parameters, which could influence the results, is essential by the application of regression analyses and should be adjusted for. Is it possible that the reason of the conflicting data regarding the association of the two parameters depends on the adjustment of the models used by the different studies? What are the other possible reasons for the conflicting data?

Minor points

  1. The authors applied different statistical tests. These tests are described in the methods section, however they are missing by the table and figure legends. It would be better to describe the applied tests in each table/figure legends. Also please identify the patient number for each data like for Table 2 to show how many patients belong to each quartile.
  2. Do the numbers shown as x axes represent linear scale in Figure 3? It seems that the odds ratios are not shown in a linear scale.
  3. The authors claim in lines 166-167 that the follow-up period between progressors and non-progressores were tended to be longer, however in Table 1 a significant p-value (p<0.001) is shown.

Author Response

On behalf of the authors, I would like to thank you for the valuable and helpful comments on our submitted manuscript, which helped improve this manuscript substantially.

We have carefully reviewed the comments and have revised the manuscript accordingly. Our responses are provided in a point-by-point manner, and changes to the manuscript are shown in red color.

Please see the attachment for our point-by-point responses to your comment.

Round 2

Reviewer 2 Report

Yun Kyung Cho and colleagues provided a revised manuscript, which shows many improvements. Nevertheless, the manuscript contains several issues.

1. In the new manuscript version the authors applied new models, which were adjusted to HDL-cholesterol (Table 2). These new models were used to calculate odds ratio, which are different from the previous odds ratios. Nevertheless, the authors did not change the text of the manuscript (Line 29, 207-217, 256, etc) according to the new odds ratio since the manuscript contains the previous odds ratio of the previous models. These mistakes should be corrected throughout the manuscript.

2. The authors calculated new p-values for Table 3, however in the manuscript text the previous p-values are reported (lines 237-238).

3. The authors included a new figure, and the figure numbers are wrong in the figure legends and also in the text, which has to be corrected throughout the manuscript.

4. The authors claim (line 166-167) that the progressors were less physically active, however the given p-value in Table 1 is non-significant.

5. The ROC curve should be referenced also in the text.

6. Typos should be corrected like in line 196 “wasompared”.

Author Response

I would like to thank you for the valuable comments on our manuscript. We deeply apologize our careless mistakes in the manuscript, and sincerely appreciate your considerate comments on our mistakes.

We have carefully reviewed the whole manuscript and have revised the manuscript according to your comments. Our responses are provided in a point-by-point manner, and changes to the manuscript are highlighted in red color.

Please see the attachment for our point-by-point responses to your comment.
